# Genetic Algorithm for Feature Selection Applied to Financial Time Series Monotonicity Prediction: Experimental Cases in Cryptocurrencies and Brazilian Assets

**DOI:** 10.3390/e26030177

**Published:** 2024-02-20

**Authors:** Rodrigo Colnago Contreras, Vitor Trevelin Xavier da Silva, Igor Trevelin Xavier da Silva, Monique Simplicio Viana, Francisco Lledo dos Santos, Rodrigo Bruno Zanin, Erico Fernandes Oliveira Martins, Rodrigo Capobianco Guido

**Affiliations:** 1Department of Computer Science and Statistics, Institute of Biosciences, Letters and Exact Sciences, São Paulo State University (UNESP), São José do Rio Preto 15054-000, SP, Brazil; guido@ieee.org; 2Department of Applied Mathematics and Statistics, Institute of Mathematical and Computer Sciences, University of São Paulo, São Carlos 13566-590, SP, Brazil; vitor.trevelin.xavier.silva@gmail.com (V.T.X.d.S.); igor.trevelin.xavier@gmail.com (I.T.X.d.S.); 3Department of Computing, Federal University of São Carlos, São Carlos 13565-905, SP, Brazil; moniquesimplicioviana@estudante.ufscar.br; 4Faculty of Architecture and Engineering, Mato Grosso State University, Cáceres 78217-900, MT, Brazil; franciscolledo@unemat.br (F.L.d.S.); rodrigo.zanin@unemat.br (R.B.Z.); profericomartins@unemat.br (E.F.O.M.)

**Keywords:** feature selection, genetic algorithm, Bitcoin, time series, forecasting, machine learning

## Abstract

Since financial assets on stock exchanges were created, investors have sought to predict their future values. Currently, cryptocurrencies are also seen as assets. Machine learning is increasingly adopted to assist and automate investments. The main objective of this paper is to make daily predictions about the movement direction of financial time series through classification models, financial time series preprocessing methods, and feature selection with genetic algorithms. The target time series are Bitcoin, Ibovespa, and Vale. The methodology of this paper includes the following steps: collecting time series of financial assets; data preprocessing; feature selection with genetic algorithms; and the training and testing of machine learning models. The results were obtained by evaluating the models with the area under the ROC curve metric. For the best prediction models for Bitcoin, Ibovespa, and Vale, values of 0.61, 0.62, and 0.58 were obtained, respectively. In conclusion, the feature selection allowed the improvement of performance in most models, and the input series in the form of percentage variation obtained a good performance, although it was composed of fewer attributes in relation to the other sets tested.

## 1. Introduction

Time series analysis is a field of research that seeks to find techniques to analyze the dependency relationship between adjacent time series observations [1]. In the context of the stock market, the analysis of market movements has resulted in two schools of thought: fundamental analysis and technical analysis, which evaluate the company’s financial structure and the price movements of its stocks, respectively [2]. Above all, regardless of the assumptions investors adopt, using time series analysis tools, which help in planning investments and managing their risks, is a common objective when analyzing the financial market [3].

In parallel to stock exchange investments, cryptocurrencies were created recently. They have great market acceptance, and have already been incorporated into the strategies of several investment funds. Furthermore, the academic community has already conducted many studies on investments in this new type of financial asset, as presented in the literature review by Fang et al. [4].

Bitcoin, known as the oldest cryptocurrency [5], is a digital currency proposed in an article published in 2008 under the pseudonym Satoshi Nakamoto, and which soon became a very attractive form of payment due to the possibility of its transactions being fast, with low costs, and almost anonymous, protecting the identity and privacy of the parties involved [6]. And so, like other world currencies, it can also be traded through an open market, with its value defined exclusively by the participants in these negotiations [7].

Cryptocurrency exchanges make buying or selling crypto assets possible, much like stock exchange assets. It is possible to mention some advantages of cryptocurrency trading: its prices are very volatile, attracting speculators, and its market is open 24 h a day and seven days a week [4]. According to coinmarketcap.com, accessed on 18 December 2023, the cryptocurrency market capitalization was 1.57 trillion dollars. The Bitcoin cryptocurrency is responsible for approximately 52% of this total.

Many companies that trade financial assets, such as financial asset managers and investment banks, are increasingly investing in artificial intelligence methods to carry out their analyses, attracted by the potential profit these emerging methods can provide [8].

The various scientific studies that have been carried out on predicting the time series of financial assets are evident [2,9,10,11,12,13,14]. According to Cho et al. [9], there are two most commonly used ways to predict the future value of these series. The first focuses on improving feature selection methods, that is, it aims to find the best set of variables capable of explaining the movement of the time series. The second aims to improve prediction models, that is, it seeks a model capable of predicting the future value of the time series with the smallest error.

In this context, two machine learning systems are proposed in this paper. The objective of both systems is to make daily predictions of the movement (up or down) of the value of financial time series. The first system will be used to predict movements in the price of the cryptocurrency Bitcoin, and the second system will be used to predict movements in the value of the Ibovespa stock index on the Brazilian stock exchange (B3), and also the price of the stocks of the company Vale, in B3. This type of prediction is very useful to assist an investor’s daily decision-making in order to serve as an indication for the eventual purchase or sale of a credit share. Specifically, if a financial series is expected to decline and, therefore, lose market value, then it is advised that it be sold before this happens. The performance of the systems will be evaluated by varying several individual requirements, which will be presented later. Thus, a comparative analysis of the best components that can make up each system will be carried out. This analysis will focus mainly on the different feature selection methods, which, as mentioned previously, is one of the most important aspects of the search for better predictions of the time series of financial assets. However, other points will also be analyzed, such as which models improve system performance and the best format for input data for these models. Therefore, the contributions of this paper will be the following:Proposal of two systems for predicting financial time series;Performance analysis focusing on feature selection methods;Performance analysis of the combination of other machine learning techniques in the systems.

The sections of this paper are organized as follows: In Section 2, the scientific studies that inspired our machine learning system are presented. Section 3 presents the theoretical foundation and proposed structure for this work, including the forms of data acquisition, analysis, and preprocessing of this data and modeling. In Section 4, the results obtained through the applied method are presented. In Section 5, the discussion of the results is presented. Finally, Section 6 concludes the study according to the objectives.

## 2. Related Works

### 2.1. Feature Selection

Generally, data with a high dimensionality of variables can be a problem for machine learning algorithms. The greater the number of variables *n* present in the data, the more evident the empty spaces become, which are the intervals in the *n*-dimensional space of the data without observations. Therefore, the training data would need to grow exponentially to ensure that these empty spaces do not exist [15]. This phenomenon is called the curse of dimensionality.

Among the methods that seek to solve this problem, feature selection has the advantage of data interpretability, which does not occur with classical methods, such as, for example, Principal Component Analysis (PCA), which subjects the data to a transformation [16]. Currently, feature selection has become an essential part of machine learning processes for data preprocessing. This is due to the increasing dimensionalities in real datasets [17].

Methods such as meta-heuristic searches are widely adopted in the feature selection process in order to obtain satisfactory results [9,11,18]. These methods, instead of finding an optimal solution to the problem, aim to obtain a reasonable solution in a shorter period of time. Furthermore, they are designed to solve a wide variety of NP-hard optimization problems, which, in general, do not have a satisfactory specific approach. In view of this, meta-heuristics are advantageous, as they do not require major adaptations to be applied to specific problems [19].

According to Dokeroglu et al. [17], feature selection methods can be classified as follows: filter, wrapper, and embedded. Filters select variables according to the value of statistical measures calculated with the data. Examples of statistical measures are information gain and correlation. Wrappers choose subsets of variables through search algorithms with a classifier while testing the performance of each subset in the predictive model. In embedded systems, the selection is carried out at the same time as the model is adjusted.

Dokeroglu et al. [17] carried out a literature review on the use of meta-heuristics for feature selection and a survey of the most applied meta-heuristics among studies from 2000 to 2022. Dokeroglu et al. [17] observed that the genetic algorithm (GA) is the most used meta-heuristic among the classic ones.

### 2.2. GA for Feature Selection

GAs are techniques that belong to the class of meta-heuristics, which are methods designed to solve optimization problems. It is a process of generating and exploring a search space using techniques generally inspired by elements of nature [20]. GAs explore the search space by imitating the biological evolution of living organisms. The process of structuring these algorithms begins with coding the solutions and defining the fitness function. Next, selection, crossover, and mutation operations must be defined. After defining the structure, the algorithm execution process begins by creating an initial random population of chromosomes. Then, for each individual, their fitness value is calculated. Soon after, the descendants of this population are generated. For this to occur, pairs of chromosomes are selected according to their fitness value and then go through the crossover process. All descendants generated go through the mutation process and, finally, replace the previous population. From that point on, the process is repeated until the specified number of generations is reached [21].

Several works used GAs to perform feature selection on financial time series data [9,11,18]. Cho et al. [9], for example, proposed the use of a GA for feature selection to predict the Korean KOSPI index and the Bitcoin cryptocurrency. Feature selection was performed using a filter and wrapper method. The filter method performed the fitness calculation with an equation composed of statistical measures such as mutual information, the F-statistic, and the Pearson correlation coefficient. The wrapper method used the accuracy of machine learning algorithms as a fitness function. The algorithms used were Support Vector Regression (SVR), Extra Trees, and Gaussian Process Regressor. Both feature selection methods showed improvement in the predictive capacity of machine learning algorithms for predicting KOSPI and Bitcoin.

Another example is the work of the authors Choudhry and Garg [11], who proposed a model for predicting the future direction of stock prices by combining a GA with a Support Vector Machine (SVM) model. For each predicted action, the main companies whose actions were highly correlated with the target variable were found, and, for each of these companies, 35 attributes were generated corresponding to the main technical indicators used by finance experts. A GA was applied to select the most important attributes, and they served as input for the SVM algorithm. The study concluded that the correlation between companies and the feature selection significantly improved the prediction performance in relation to the use of the SVM model alone.

In the study by Tsai and Hsiao [18], they proposed the combination of three different feature selection methods for stock price prediction. The selected attribute subsets were generated from the intersection of results from the PCA, GA, and Decision Trees (DT) methods. The experiments evaluated the prediction performance using an Artificial Neural Network (ANN). From the results obtained, the benefit of combining different feature selection methods was clear.

### 2.3. Data Types and Preprocessing for Financial Time Series

Many works that study financial time series propose the use of different types of data in their machine learning systems. Some studies calculate technical analysis indicators [12], others use technical analysis data in conjunction with fundamental analysis [2,13,14], other studies propose the preprocessing of time series such as the use of lag [10], and others use real and percentage changes [9].

Ayala et al. [12], for example, propose a hybrid method that makes use of two technical analysis indicators: the Triple Exponential Moving Average (TEMA) and the Moving Average Convergence/Divergence (MACD). Their study trained machine learning models: Linear Regression, ANN, Random Forests (RF), and SVR with data relating to the Open, High, Low, and daily Close of indices, such as the DAX. They then combined the models’ prediction with the indicators’ buy and sell signals into a hybrid model and concluded that it was able to generate more profit and reduce investment risk.

In contrast, Akita et al. [13] propose the use of news and financial time series of various stocks. Akita et al. [13] used a Paragraph Vector to generate a vector representation for each of the news items. For time series data, they used a combination of daily values of the Open, High, Low, and Close data for each of the stocks studied. The results showed that the use of textual news data generated gains in predictive performance when compared to the use of only numerical data from financial time series. Akita et al. [13] also demonstrated that there are gains in predicting the future price of a company when data from other companies in the same line of business are used to train the models.

The study carried out by Picasso et al. [2] combined technical and fundamental analysis indicators to make predictions on the Nasdaq 100 index. From the data on the daily values of this index, ten numerical data relating to technical analysis indicators were calculated, such as, for example, the MACD, the Simple Moving Average (SMA), the Exponential Moving Average (EMA), and the Relative Strength Index (RSI). Picasso et al. [2] also extracted news data from the financial market to perform a sentiment analysis of this textual data.

External factors, such as public sentiment and political events, can also impact stock price fluctuations. This phenomenon was studied by Khan et al. [14], who extracted and processed data from Twitter posts, political news from Wikipedia, and time series from Yahoo! Finance. In the end, Khan et al. [14] applied several machine learning models to the data and concluded that public sentiment data can improve its predictive capacity by up to 3%, while political events generate an improvement of up to 20%.

Cho et al. [9] compared two feature selection algorithms, a filter and a wrapper, using GAs to predict the KOPSI index and Bitcoin. Cho et al. [9] used a total of 264 attributes as inputs to the system, including real values and real and daily percentage changes of time series related to global economic indices, exchange rates, and commodities. As a result of the experiments, Cho et al. [9] concluded that the percentage changes of the series was the best type of input data for model learning.

To predict the movement of Bitcoin prices, Mallqui and Fernandes [10] used a time series related to Bitcoin prices, internal data from the Bitcoin network, and also index values and commodities. As preprocessing, lags, and smoothing by Weighted Moving Average (WMA) were applied to some of these time series. As a result, the models that were trained with this data were able to achieve better performances than those presented in the state-of-the-art models for predicting the direction of Bitcoin’s price.

### 2.4. Machine Learning in Financial Time Series Prediction

The final stage of the machine learning process is training predictive models and verifying the results obtained.

To predict stock prices, many studies use statistical models for time series analysis, such as the Autoregressive Conditional Heteroscedasticity (ARCH), the Autoregressive Moving Average (ARMA), and the Autoregressive Integrated Moving Average (ARIMA) models [22]. However, artificial intelligence became popular and, consequently, so did machine learning. These new technologies prove to be more effective than statistical models in making predictions on noisy data, such as, for example, financial time series [23].

The literature review carried out by Kumbure et al. [22] selected the most relevant articles from 2000 to 2019 on machine learning and types of data for stock market predictions. It was found that most articles used regression models and classification models. To perform regression and classification tasks, researchers mainly used Neural Networks and Support Vector Machine (SVM) models. However, deep learning models, ensembles, KNN, and others were also found.

### 2.5. Final Considerations

In this section, several literature studies that address the prediction of financial time series were presented. Each of these studies contributes, in a unique way, to the four main topics of this study’s problem: the collected data, the preprocessing of the series data, the feature selection, and the machine learning models. However, as noted, the financial time series prediction problem is complex, and there is no consensus on the best methods to solve it. In summary, each financial time series has its own particularities. Therefore, to predict it, a more specific approach may be necessary.

To build the two systems proposed in this paper, we chose to test some of the strategies found in the literature related to the four main topics to approach the problem, which were mentioned above. Various combinations of these methods will be tested and compared, aiming to predict the financial time series targeted in this work.

## 3. Materials and Methods

In this section, the proposed methodology will be described, and each of the steps will be detailed. The objective of this work, as already mentioned in Section 1, is to make daily predictions about the monotonicity, that is, the direction of movement, of the value of financial time series through two systems. For example, at the beginning of a trading session for stock exchanges, or at the beginning of the day for cryptocurrencies, an attempt was made to predict whether the values of these series would have a positive or negative variation at the end of the same day. Mathematically, given a financial time series *s* defined as s:N→R, our goal is, in fact, to perform predictions and analyses on the time series *g* defined in Equation (Equation 1):(1)$$\begin{array}{ccccc} g & : & N & \to & \left\{0,1\right\}\\ \; & \; & t & \mapsto & \mathrm{signal}\left(s(t)-s(t-1)\right), \end{array}$$
in which the function signal is defined as:(2)$$\mathrm{signal}\left(x\right)=\left\{\begin{array}{cc} 1, & \mathrm{if}\;x>=0,\\ 0, & \mathrm{if}\;x<0. \end{array}\right.$$

These systems will consist of time series preprocessing methods, feature selection with GAs, and classification models. The systems will be applied to the time series of Bitcoin, Ibovespa, and Vale stocks.

The proposed methodology can be seen in Figure 1, below, which summarizes the following steps: data collection, data preprocessing, data partitioning between training and testing, feature selection, training of each model, and the evaluation of the models.

The methodology proposed in this work is an adaptation of several points from the study by Mallqui and Fernandes [10], who carried out experiments to predict the value of Bitcoin, and their results are compared and evaluated with the performance of the framework proposed in this work in the Bitcoin scenario. The adaptations of the study by Mallqui and Fernandes [10] that were necessary to apply the approach to the Ibovespa and Vale stock series are also detailed in this section.

To facilitate understanding throughout the work, the first system, applied to the Bitcoin series, will be referred to as Experiment 1 (E1), and the second system, applied to the Ibovespa and Vale series, will be referred to as Experiment 2 (E2) and Experiment 3 (E3), respectively.

### 3.1. Data Collection

Data from several time series relating to the historical prices of financial assets were collected to conduct the experiments. These data referred to the values of various company stocks, commodities, stock indices, futures contracts, and exchange rates. This broad collection of data external to the target time series was motivated by the study by Cho et al. [9]. The purpose was to choose, through feature selection algorithms, only those most relevant data for adjusting machine learning models.

To compose the initial dataset, the daily time series of several financial assets were considered. Each asset has five series: Open, High, Low, Close, and the trading volume for each trading day. All time series, with the exception of the Bitcoin series, were collected from the https://finance.yahoo.com/ Yahoo Finance website. For Bitcoin, as in the work of Mallqui and Fernandes [10], internal data from the cryptocurrency was extracted, and, in addition to the five series mentioned above, transaction fees, the cost per transaction, and the Bitcoin hash rate were also used. These data were extracted from two sources: the series relating to Bitcoin prices was obtained from the https://www.investing.com/crypto/bitcoin Investing.com website, and the rest of the series relating to Bitcoin was collected from the https://data.nasdaq.com/ Nasdaq Data Link website, accessed on February 2, 2024.

Table 1 below refers to E1, and Table 2 refers to E2 and E3, and they present the input time series for each experiment.

It is worth highlighting that the intention of this work was precisely to optimally select the set of temporal variables that accompany the main financial series established in each of the experiments considered, in order to reduce the forecasting error for future values of these series. Therefore, the complete space of features that we considered was formed by the series in Table 3, Table 4, and Table A1 of the Appendix A, but our objective was to select a partition from these sets in order to enhance the performance of the models considered. Therefore, in each execution of our framework, we had a specific subset of features.

### 3.2. Data Preprocessing

The preprocessing phase was carried out after data collection. At this stage, the data were prepared to be used as input for the models. In Section 2.3, we saw that each considered work presented a different way of preprocessing the analyzed time series data. However, they all had the same objective, which was to generate new series that were more appropriate for effective learning in the training phase.

For E1, Bitcoin, some attribute engineering techniques proposed by Laboissiere et al. [3] were applied to the collected data, such as the concept of lag and smoothing of time series. Table 3 lists all the attributes of the dataset already processed for E1. In this table, the day D column represents the day for which we wanted to predict the direction. For this day, only the Open value was available. The Lag (1 to 7) column represents a series with a delay of 1 to 7 days. Finally, the WMA column (30 days) represents the series for which the Weighted Moving Average indicator with a period equal to 30 days was calculated.

As seen in Section 2.3, there are many different ways to preprocess time series data. Cho et al. [9], for example, contributed to the identification of daily percentage changes as an optimal format for financial time series, which were used as input in machine learning models. On the other hand, Mallqui and Fernandes [10] used the concept of lags, that is, variables that represent periods in the past or the delay of a time series.

In this sense, the second system, referring to E2 and E3, also had its models subjected to different types of data, with four different datasets in total. These sets were created from the assets presented in Table 2. The first dataset (D1) is the same as that proposed by Mallqui and Fernandes [10], with the addition of a financial series and adaptation to the target series analyzed. Table 4 specifies the composition of the D1 dataset. As in the E1 dataset, the preprocessing carried out on the data from this experiment involved the application of lags and smoothing of the time series with WMA. More details on the basis of time series smoothing and preprocessing methods can be found in the work of Ranjan et al. [24].

In the second dataset (D2), the Open, High, Low, and Close series of the target asset were selected. Meanwhile, for external assets, only the Close series were selected. For each of these series, in addition to its original value, two more variations were added, which represent the real difference and percentage difference of the series with itself in the previous period (lag of 1 period), as done in Cho et al. [9]. Table 2 presents the financial assets whose series were used to calculate the D2 attributes for E2 and E3.

To compose the third dataset (D3), the conclusion of the study by Cho et al. [9] was used: the representation of percentage changes of the series surpasses both its original form and the real changes in the performance of machine learning models. Therefore, D3 was composed only of the percentage change data present in D2. Finally, the fourth dataset (D4) was similar to the set proposed by Yun et al. [25]. For each experiment, the five series of the target asset were used, which are the Open, High, Low, Close, and trading volume series. Furthermore, several technical analysis indicators were calculated for these five series. The complete composition of D4 is specified in Table A1 of Appendix A.

Filling in missing data, using the last valid value in the series, was carried out before applying any transformation to the original data. These missing values were identified in the datasets and represent non-trading days, such as weekends and holidays. Assuming that the missing data in the series were mostly caused by non-trading days, we chose to handle them by replacing the missing values with the last valid observation in the series. This approach was chosen over others for its simplicity and ease of implementation. Furthermore, it is worth mentioning that we used this procedure in all time series involved in the experiments covered in this paper. It is worth highlighting that the choice of replacing missing values with the last valid value in the time series was taken due to its simplicity and the characteristics of the implemented systems. The Bitcoin system did not need sophisticated imputation considering that series with missing data were only used after being transformed into the 30-day WMA and that they represented only a small portion of the system’s total input series. In the case of the second system, referring to the Ibovespa and Vale series, again, there were a few series that presented missing values, mainly because the series used were, in most cases, from the Brazilian Stock Exchange itself, B3. As there were very few missing values, a more sophisticated imputation method would hardly have yielded much better results, so we opted for this simpler method.

Before the data was used to train the models, it was normalized so that its values fell within the range [0,1]. For each of the experiments (E1, E2, and E3), the target variable took on two different values: 1 for days when the daily variation was positive and 0 otherwise.

### 3.3. Data Partitioning

After preprocessing, the data from each experiment was divided into two intervals. Then, each interval was divided into a training set and a testing set so that it was possible to evaluate the performance of the machine learning models. Considering the data that were collected for each experiment were time series, in which each observation has a dependent relationship with the next, the series chronological order must be preserved. Thus, the training and test set should contain only data on the initial and final period of the interval, respectively, without overlap. Furthermore, in E1, these intervals represented exactly the same periods used in the study by Mallqui and Fernandes [10], allowing a comparison between the results obtained. The interval I2 was a larger dataset, proposed by Mallqui and Fernandes [10] as a baseline for future research. Thus, in our experiments, the dataset from the interval I2 was larger than I1. Table 5 presents the start date, the end date, the periods, and the partition sizes for the training and testing sets for each interval of this work.

### 3.4. Feature Selection

Several attributes were generated by the preprocessing carried out. Some of the datasets had more than a hundred attributes, but many may be irrelevant or even redundant to solve the problem, which can become unduly complex. Furthermore, data with the high dimensionality of attributes may fall under the curse of dimensionality, a phenomenon explained in Section 2.1, which would harm the performance of machine learning algorithms. To avoid this induction and reduce complexity, a feature selection was carried out.

The filter and wrapper methods were applied to promote this feature selection, which was driven by a GA. This search algorithm was presented in Section 2.2, and below, its structure within the scope of this work is defined.

#### 3.4.1. Solution Coding and Fitness

In a GA, the solution is encoded as a sequence of bits. Each bit of this sequence is related to an attribute of the dataset. This way, the selected attributes are represented by bits with value 1 in the solution. On the other hand, bits with a value of 0 represent attributes that are discarded.

Fitness values are obtained by calculating fitness functions over the solutions. However, this calculation is performed differently depending on the approach. For filter methods, the fitness function is calculated as:(3)$$fitness=\sum_{i=1}^{n}f_{S_{target},S_{i}}-\sum_{i=1}^{n-1}\sum_{j=i+1}^{n}f_{S_{i},S_{j}},$$
in which $f_{S_{i},S_{j}}=MI_{S_{i},S_{j}}+F_{S_{i},S_{j}}+C_{S_{i},S_{j}}$; *n* corresponds to the number of attributes selected by the solution; $S_{target}$ is the target variable; and MI, *F*, and *C* refer to measures of mutual information, *F*-statistics, and Pearson correlation coefficient, respectively. This fitness calculation function was introduced by Cho et al. [9] and includes a search for subsets of attributes that are highly correlated with the target variable while, at the same time, having little correlation with each other. It is worth mentioning that, according to Kraskov et al. [26], mutual information represents the relationship between two random variables and, unlike the correlation coefficient, it is sensitive to dependencies not evident in covariance.

The value of mutual information between two variables *x* and *y* is defined by: (4)$$I\left(x;y\right)=\sum_{i=1}^{n}\sum_{j=1}^{n}p\left(x\left(i\right),y\left(j\right)\right)\cdot \log\frac{p(x(i),y(j))}{p(x(i))\cdot p(y(j))},$$
whose value is equal to 0 when *x* and *y* are statistically independent [26]. Its value can be used in the context of attribute selection, allowing it to quantify the relevance of a certain subset of variables in relation to the target vector.

The Pearson correlation coefficient is a statistical measure used to measure the strength and direction of the linear relationship between two variables [27]. The Pearson correlation coefficient between two variables *x* and *y* is defined by: (5)$$\rho_{xy}=\frac{\sum \left(x_{i}-\bar{x}\right)\sum \left(y_{i}-\bar{y}\right)}{\sqrt{\sum {\left(x_{i}-\bar{x}\right)}^{2}}\sqrt{\sum {\left(y_{i}-\bar{y}\right)}^{2}}},$$
where $\bar{x}=\sum_{i=1}^{N}x_{i}$, $\bar{y}=\sum_{i=1}^{N}y_{i}$, and $\rho_{xy}$ has values between −1 and 1. $\rho_{xy}=0$ is an indication that the variables are not correlated and the correlation is stronger as $|\rho_{xy}|$ approaches 1.

The F-statistic measure used in the formula refers to the significance value of the F-test [28]. The F-test is a statistical hypothesis test used to determine if the difference in variance between two samples is significant. We will adopt the F-statistic of this test in the filter method fitness formula to quantify this relationship between each of the selected features. The bigger its value, the more significant the relationship between the features.

On the other hand, for the wrapper method, the calculation of the fitness value involves training a learning model and evaluating the performance of that model. To achieve this, the training set for each interval, illustrated in Table 5, was divided into two parts, 80% and 20% for the interval I1, and 75% and 25% for the interval I2. The first part was intended for training the model with the attributes selected by the GA solution. The second part was used to test the performance of this trained model. Based on the predictions made in the model test, the fitness of the solution was calculated as the value of the area under the ROC curve (AUROC) metric. It is important to note that, although the GA fitness function is defined based on the ROC AUC metric, we also considered the accuracy of the model in our experiments.

#### 3.4.2. Operators and Parameters

In this work, the essential operators of a GA were adopted: selection, crossover, and mutation. Furthermore, elitism was used.

Among the parameters that control the execution of the GA, the number of generations, the population size, and the mutation rate were defined. The population size and the number of generations parameters represent, respectively, the number of chromosomes present in each generation and the number of times the process of generating new offspring is repeated. Table 6 and Table 7 show the values defined for each of these parameters, as well as a summary of the operators for E1, E2, and E3.

The selection operator chooses the most fit chromosomes for the reproduction and generation of the next offspring. The selection operator is defined, depending on the experiment, such as with the Roulette Wheel or Rank methods. In the Roulette Wheel method, individuals in the population are chosen to be part of the next with a probability proportional to their respective fitness, having the best individuals (larger fitness) having a greater likelihood of being chosen. In the Rank method, individuals are ranked according to fitness and their rank position determines their chance to be chosen [21].

The operator crossover randomly chooses a point to divide the sequence of two chromosomes. The initial part of the sequence of a chromosome will come together with the final part of the other, generating a new chromosome. To perform the crossover, the method of two points was used in all experiments, so chromosomes were divided into two distinct points. The mutation operator may randomly change the value of any chromosome gene but with a small probability. And, as explained in Section 2.2, after the mutation, the descendants replace the old population. In this process, we used the elitism method, which forces the GA to keep part of the best chromosomes in the next generation [21].

### 3.5. Model Training

In E1, a performance comparison of different types of machine learning algorithms was carried out: Support Vector Machine (SVM), Random Forest (RF), Multilayer Perceptron (MLP), K-Nearest Neighbors (KNN) and Logistic Regression (LR).

For E2 and E3, a comparison was made between three models: the Support Vector Machine (SVM), the Decision Tree (DT), and the K-Nearest Neighbors (KNN) models.

In all experiments, the models were subjected to a hyperparameter optimization process, which was carried out after the feature selection process and before the models made predictions on the test set. This process was carried out using the Grid Search method.

Grid Search is the simplest method to perform hyperparameter optimization. This method works as follows: the user specifies a finite set of values for each hyperparameter, and the Grid Search algorithm evaluates the Cartesian product of these sets, that is, it tests all possible combinations between the hyperparameter values. In the end, the best combination of parameters found is adopted in the system [29].

To carry out this process and define the best hyperparameters, the training set was divided into two partitions, 80% and 20% for the I1 interval, and 75% and 25% for the I2 interval. The models were trained on the first partition, and the second partition was used to calculate the area under the ROC curve (AUROC). In the end, the hyperparameter set that obtained the best value for this metric was selected.

It is worth mentioning that in the wrapper method, the models were applied to calculate fitness. This way, for each model, default hyperparameter values were defined so that they could be used in this step. The sets of hyperparameters evaluated in E1 in the Support Vector Machine (SVM), Random Forest (RF), Multilayer Perceptron (MLP), K-Nearest Neighbors (KNN), and Logistic Regression (LR) models are summarized in Table 8. The sets of hyperparameters evaluated in E2 and E3 in the Support Vector Machine (SVM), Decision Tree (AD), and K-Nearest Neighbors (KNN) models are summarized in Table 9.

Given the operation of the proposed method, we can highlight some points that represent the main differences when compared to the works of Cho et al. [9] and Mallqui and Fernandes [10], which served as inspiration for our developments. Specifically, the work of Cho et al. [9] introduced the concept of cuts and genetic filters for predicting values of financial time series with SVR. In contrast, in this work, we used these feature selection approaches to predict movements of appreciation, stagnation, and devaluation in time series using SVM, RF, MLP, KNN, and LR. Regarding the work of Mallqui and Fernandes [10], our advances share similarities in the objective, which is to predict the monotonicity of the financial time series by analyzing various classifiers. However, there are differences in the strategies of feature selection, as the authors used dimensionality reduction through PCA and correlation analysis. Furthermore, none of these works consider, as done in this text, time series of Brazilian assets, which are recognized as challenging in prediction tasks [30].

### 3.6. Model Evaluation

It is important to note that the central objective of this work is to predict monotonic behaviors, that is, the increase and decrease, of time series. Therefore, proceeded in the same way as Mallqui and Fernandes [10], who only considered metrics related to the accuracy of the classification models. Specifically, the accuracy and area under the ROC (AUROC) curve metrics were used to evaluate the performance of each of the models, which were trained to predict the direction of financial asset prices in each of the three experiments. These metrics were calculated based on the predictions made on the test sets. It is worth mentioning that each of the proposed models were trained and tested fifty times to obtain statistically significant values for each of the calculated metrics.

The ROC curve is a graph that plots the false positive rate (FPR) on the *y* axis and the true positive rate (TPR) on the *x* axis [31]. The calculation of these parameters is represented in Equations (Equation 6) and (Equation 7).
(6)$$\mathrm{FPR}=\frac{\mathrm{FP}}{\mathrm{FP}\;+\;\mathrm{TN}}.$$
(7)$$\mathrm{TPR}=\frac{\mathrm{TP}}{\mathrm{TP}+\mathrm{FN}}.$$

The AUROC is a metric that summarizes the ROC by calculating the area over the curve, with its value varying from 0.0 to 1.0, and the higher it is, the better the model’s classification performance. In addition to AUROC, accuracy was considered as a comparison parameter. Accuracy is calculated using the following expression:(8)$$\mathrm{Accuracy}=\frac{\mathrm{TP}+\mathrm{TN}}{\mathrm{TP}+\mathrm{FP}+\mathrm{TN}+\mathrm{FN}}\cdot 100\%.$$

## 4. Results

The systems presented in Section 3 were applied, and the classifier models resulting from each experiment went through a process of validation of their performance. The best results obtained from this validation are presented in this Section. Tables 10, 12 and 13 show the results of E1, E2, and E3, respectively.

### 4.1. Missing Data

Before running the experiments, we accounted for missing data in each dataset. The amount of missing data in each feature of the datasets are listed below:E1 (Bitcoin): KOSPI (19.10%), Platinum (18.17%), Palladium (18.17%), Coffee (18.17%), Copper (18.12%), Oat (18.12%), Silver (18.12%), Sugar (18.12%), NASDAQ (18.12%), Natural Gas (18.12%), Heating Oil (18.12%), Gold (18.12%), S&P500 (18.12%), Crude Oil (18.12%), Cocoa (18.12%), and DAX (18.06%);E2 (Ibovespa): Crude Oil (2.33%), E-Mini S&P 500 (2.33%), Gold (2.43%), Copper (2.43%), Wheat (2.33%), MSCI (2.53%), Natural Gas (2.33%), Nasdaq 100 (2.33%), Silver (2.43%), Corn (2.53%), Oat Futures, May-2023 (2.33%), Rough Rice Futures, May-2023 (3.54%), Soybean (2.33%), Cboe UK 100 (2.02%), Dow Jones (2.33%), FTSE 100 (2.23%), DAX (1.21%), S&P 500 (2.33%), NASDAQ (2.33%), NYSE (2.33%), Russell 2000 (2.33%), NYSE AMEX (2.33%), Euro (0.10%), and US Dollar (0.10%);E3 (Vale): Crude Oil (2.33%), E-Mini S&P 500 (2.33%) Gold (2.43%), Copper (2.43%), Wheat (2.33%), MSCI (2.53%), Natural Gas (2.33%), Nasdaq 100 (2.33%), Silver (2.43%), Corn (2.53%), Oat Futures, May-2023 (2.33%), Rough Rice Futures, May-2023 (3.54%), Soybean (2.33%), Cboe UK 100 (2.02%), Ibovespa (0.10%), Dow Jones (2.33%), FTSE 100 (2.22%), DAX (1.21%), S&P 500 (2.33%), NASDAQ (2.33%), NYSE (2.33%), Russell 2000 (2.33%), NYSE AMEX (2.33%), Euro (0.10%), and US Dollar (0.10%).

### 4.2. E1—Bitcoin

After carrying out the experiments, the classification models were evaluated, and those that presented the best results in relation to the AUROC and accuracy metrics are presented in Table 10 for both intervals. Experiments that were performed without applying feature selection have the value “none”. From these values, we can observe that for the first interval, the best results were obtained with the application of filter and wrapper methods for feature selection. For the second interval, the experiments that resulted in better values for the AUROC were those in which feature selection was not performed, that is, all the input features from the dataset were used.

Analyzing the results presented in Table 10, it can be observed that the ANN and RF models did not demonstrate good performance in both intervals, resulting in AUROC values very close to 0.5, that is, their performance was similar to that of a random classifier.

Observing the results for interval 1, it is clear that the SVM model presented the best results, with a AUROC of 0.6 and an accuracy of 59.81%. In this case, the 25 attributes selected by the filter feature selection were used. The attributes used are indicated in Table 11. It is interesting to highlight the selected variables referring to the Weighted Moving Averages of the Coffee close series and Platinum close series, indicating a possible correlation between the price trends of these commodities and the price movements of Bitcoin.

### 4.3. E2 and E3—Ibovespa and Vale

For the Ibovespa and Vale experiments, the best results per model and interval are presented in Table 12 and Table 13. These results were also chosen by the highest AUROC value. The rows in the tables represent each of the machine learning models applied in these experiments: Support Vector Machine (SVM), Decision Tree (AD), and K-Nearest Neighbors (KNN). For each of these three models, there are two columns that represent the performance metrics calculated on their predictions. The metrics are accuracy (ACC) and AUROC. The Model column represents the dataset (D1, D2, D3, or D4) in which it performed best, the best feature selection method (filter, wrapper, or ’none’ selection method), and the best hyperparameters found for the same.

Each model has its own hyperparameters, as presented in Section 3.4.2. So, for the SVM model, the hyperparameters are represented as C|k|γ|d, where *C* represents the regularization hyperparameter, *k* represents the kernel function used, which can be ’r’ for radial basis or ’p’ for polynomial. γ represents the gamma parameter, and *d* is the degree of the polynomial (used only for polynomial kernels). For the DT model, hyperparameters are represented as c|p|s|f, where *c* represents the division criterion, which assumes ’g’ for Gini or ’e’ for entropy; *p* represents the maximum depth of the tree; the *s* represents the minimum number of samples to split a node; and *f*, the minimum number of samples to create a leaf node. Finally, for the KNN model, the hyperparameters are represented as k|w, where *k* represents the number of neighbors considered for classification and *w* represents the weight of each neighbor in the classification, which assumes ’u’ for uniform weight or ’d’ to consider distance as weight.

In Table 12, the best-performing models for intervals I1 and I2 in E2 are presented. The best result obtained in I1 was 0.62 AUROC and 59.86% ACC. This result was obtained by the SVM model, with feature selection using the filter method on the D1 dataset. For I2, the best model was the Decision Tree (DA), which obtained 0.61 AUROC and 61.44% ACC. This result was achieved using the D3 dataset without applying any type of feature selection.

In Table 13, the best-performing models for intervals I1 and I2 in E3 are presented. The best result obtained in I1 was 0.57 AUROC and 56.34% ACC. This result was obtained by the KNN model with feature selection using the wrapper method on the D2 dataset. For I2, the best model was the Decision Tree (DT), which obtained 0.58 AUROC and 57.41% ACC. This result was achieved using the D2 dataset without applying any type of feature selection.

## 5. Discussion

Among all the results, the Ibovespa and Bitcoin predictions were the ones that presented the best results, 0.62 and 0.61 of AUROC, respectively. Following that, the best results came from predictions for Vale’s stocks.

In this work, in Bitcoin E1, we adopted intervals equal to the work of Mallqui and Fernandes [10] to allow a comparison of results. For the I1 interval, our best AUROC result was 0.6, with an accuracy of 59.81%, in the SVM model and with filter feature selection. In the I2 interval, the best result was AUROC was 0.61, with an accuracy of 55.45%, in the KNN model and without any feature selection. In the I2 interval, it is also worth highlighting the AUROC result of 0.58 and accuracy of 65.91% by the SVM model without feature selection. In comparison, the results of Mallqui and Fernandes [10] were as follows: in I1, AUROC was 0.58, with an accuracy of 62.91%, with Ensemble A (proposed by the study) and feature selection by correlation; in I2, AUROC was 0.58, with an accuracy of 59.45%, with the SVM model, without feature selection. Our models achieved AUROC values higher than those of Mallqui and Fernandes [10]; however, not all accuracy values were equally superior.

Regarding feature selection in E1, the predominance of better results from the filter strategy in interval 1 is evident. In interval 2, it is clear that the best results were obtained without feature selection, with the exception of ANN and LR, which achieved superior performance (higher AUROC) with the wrapper strategy. In general, the application of the proposed feature selection methods presented significant results for most models, resulting in an increase in the AUROC and, in some cases, an increase in accuracy.

It is important to highlight that the methodology used in E2 and E3 were also based on the study by Mallqui and Fernandes [10], which proposed a system just for Bitcoin prediction. However, it is evident that this system, with the adaptations proposed in this work, was capable of learning and also predicting the financial time series relating to Ibovespa and Vale stocks with good performance.

For the system proposed for the Ibovespa and Vale experiments, the adaptations applied in this work include the feature selection, the different types of input data for the machine learning models, as well as the use of other models. In the end, several characteristics were tested in each resulting model. Table 14, Table 15, and Table 16 present the best results, selected by AUROC, for E2 and E3 and intervals I1 and I2. These experiments and intervals are represented in the lines and the highest AUROC value obtained for each experiment–interval pair is highlighted in bold. Table 14 presents the best results seen from the perspective of each dataset (D1, D2, D3, and D4), Table 15, for each type of feature selection (filter, wrapper, and none), and Table 16 for each model of machine learning used (SVM, DT, and KNN).

In Table 14, it is apparent that sets D1, D2, and D3 were those that performed the best. The D1 dataset was generated by the preprocessing proposed by Mallqui and Fernandes [10] with some adaptations. D1 is responsible for the highest AUROC value (0.62), which occurred in the Ibovespa experiment (E2). This shows that the processing carried out on the data was also efficient in this scenario. As for sets D2 and D3, whose processing involved transforming the original data into real and percentage changes, they were the ones that induced the best performance in most experiments. It is important to highlight set D3, which had a third of the number of attributes of set D2, and, even so, was able to surpass it in some cases.

On the other hand, the D4 set did not obtain better AUROC values in any of the scenarios covered, as shown in Table 14. Nevertheless, D4 did not have the worst value in any of these scenarios. Despite this, its performances were only relevant in the Ibovespa prediction (E2).

Table 15 shows that feature selection methods did not improve performance in all experiments, since in half of the scenarios covered (E2–I2, E3–I1, and E3–I2), the non-use of feature selection methods variables (column ’none’) reached higher AUROC values. It is worth noting that there were cases in which the GA, in the first generations, stopped finding new better solutions, and, therefore, maintained the same solution until the end of the process. Even so, in cases where the filter and wrapper methods did not improve the models’ performance (E2–I2, E3–I1, and E3–I2), the values were close to those obtained by models trained with the complete dataset, i.e., without selection.

In Table 16, it is clear that the Decision Tree (DT) model performed the best in E2 and E3.

## 6. Conclusions

In this work, the creation of two machine learning systems were proposed to predict movements in the price time series of Bitcoin, Ibovespa, and Vale stocks. The prediction of these three financial time series was studied with a focus on feature selection, but other aspects, such as preprocessing and modeling, were also evaluated. It is important to highlight that, in each of these aspects, several strategies were tested and compared using AUROC and accuracy metrics.

In the context of E1, it is clear that the use of GAs in the filter feature selection strategy demonstrated good results in the experiments in interval 1, with emphasis on the selected attributes, which were mostly extracted from the Blockchain of Bitcoin, with the exception of the WMA (30 days) of the close prices series of Coffee, Platinum, and S&P 500, indicating a possible correlation and information gain in the use of these attributes in predicting Bitcoin movements.

Among the results obtained, in relation to the datasets used in E2 and E3, dataset D3 stands out. It is composed of the daily percentage changes of the Open, Low, High, and Close series of the target asset, in addition to the percentage changes of the Close series of all financial assets destined for E2 and E3, listed in Table 2. The D3 obtained great results, and its difference is that it is composed of many fewer attributes than other datasets.

Regarding feature selection in E1 and E2, filters and wrappers based on GAs were successfully applied. However, it is important to highlight particularities regarding their use, especially with regard to the choice of the fitness function of the filters. Furthermore, in half of the scenarios presented in Table 12, it was observed that machine learning models learned better from complete datasets, indicating that more refined data preprocessing should be carried out before feature selection.

In reference to the machine learning models applied in this work, the Decision Tree (DA) stood out in the Ibovespa and Vale experiments, obtaining results, measured by the area under the ROC curve, better than the other models in half of the scenarios, as shown in Table 16.

In view of the conclusion of this work, it is important to look for new ways of preprocessing financial time series data in order to improve both the feature selection and the performance of the models. The form of preprocessing applied to the D3 dataset, which was highlighted in this study, is also a way of normalizing the data. From this perspective, it is suggested to search for other ways of preprocessing data for financial time series.

In future developments, we intend to modify our material in order to make it effective in detecting values in financial time series and not just in detecting monotonicity. Furthermore, more elaborate metaheuristics, such as those based on massive local search [32,33,34,35,36,37], should be considered in feature selection.

## Figures and Tables

**Figure 1 entropy-26-00177-f001:**
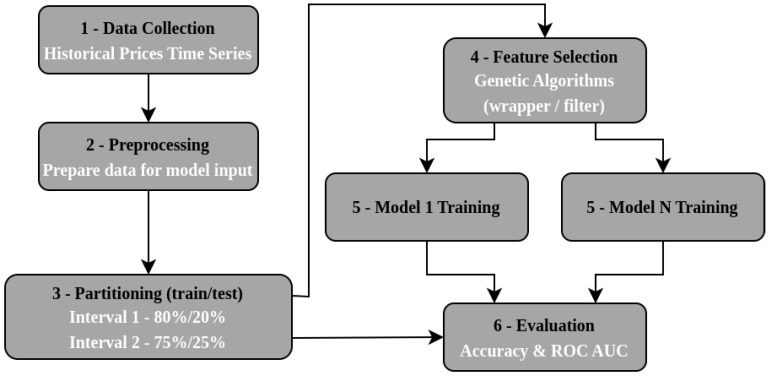
Overview of the proposed methodology.

**Table 1 entropy-26-00177-t001:** Input time series for E1. The columns show the type of financial time series and what it describes.

Type	Time Series
Cryptocurrencies	Bitcoin
Commodities	Crude Oil, Gold, Silver, Coffee, Heating Oil, Natural Gas, Platinum, Palladium, Copper, Cocoa, Sugar, Oat
Stock Market Indexes	S&P 500, NASDAQ, DAX, KOSPI

**Table 2 entropy-26-00177-t002:** Input time series for E2 and E3. The columns show the type of financial time series and what it describes.

Type	Time Series
Stocks	Vale, Petrobras, Usiminas, EcoRodovias, Raia Drogasil, Equatorial Energia, JBS, SABESP, BB Seguridade, EDP, Banco do Brasil, Itaúsa, Banco Bradesco, Multiplan, Itaú, Eletrobrás, B3, Ambev, Klabin
Commodities	Crude Oil, Gold, Rice, Soybean, Natural Gas, Silver, Corn, Wheat, Oat, Copper
Stock Market Indexes	Ibovespa, S&P 500, Dow Jones Industrial Average, NASDAQ, NYSE COMPOSITE, NYSE AMEX COMPOSITE INDEX, Cboe UK 100, Russell 2000
Futures	Nasdaq 100, E-Mini S&P 500, MSCI World Index Futures
Currencies	US Dollar, Euro, British Pound

**Table 3 entropy-26-00177-t003:** The following attributes of E1 are presented: the feature for which we will forecast monotonicity, the time series with a delay of 1 to 7 days, and the series whose WMA indicator with 30 days was calculated.

Day D	Lag (1 to 7)	WMA (30 days)
Open	Direction	Open
	Open	High
	High	Low
	Low	Close
	Close	Volume
	Volume	Number of Transactions
	Number of Transactions	Transaction Fees
	Transaction Fees	Cost per Transaction
	Cost per Transaction	Average Hash Rate
	Average Hash Rate	$C_{\mathrm{Crude}\;\mathrm{Oil}}$
		$C_{\mathrm{Gold}}$
		$C_{\mathrm{Silver}}$
		$C_{\mathrm{Coffee}}$
		$C_{\mathrm{Heating}\;\mathrm{Oil}}$
		$C_{\mathrm{Natural}\;\mathrm{Gas}}$
		$C_{\mathrm{Platinum}}$
		$C_{\mathrm{Palladium}}$
		$C_{\mathrm{Copper}}$
		$C_{\mathrm{Cocoa}}$
		$C_{\mathrm{Sugar}}$
		$C_{\mathrm{Oat}}$
		$C_{S\&P\;500}$
		$C_{\mathrm{Nasdaq}}$
		$C_{\mathrm{DAX}}$
		$C_{\mathrm{KOSPI}}$

C—close series.

**Table 4 entropy-26-00177-t004:** The following attributes for dataset D1 (E2 and E3) are presented: the feature for which we will forecast monotonicity, the time series with a delay of 1 to 7 days, and the series whose WMA indicator with 30 days was calculated.

Day D	Lag (1 to 7)	WMA (30 days)	WMA (30 days)
Open	Direction	Open	$C_{\mathrm{DAX}}$
	Open	High	$C_{\mathrm{NASDAQ}}$
	High	Low	$C_{S\&P\;500}$
	Low	Close	$C_{\mathrm{Nasdaq}\;100}$
	Close	Volume	$C_{\mathrm{Ibovespa}}$
	Volume	$C_{\mathrm{Crude}\;\mathrm{Oil}}$	$C_{\mathrm{MSCI}}$
		$C_{\mathrm{Gold}}$	$C_{\mathrm{FTSE}\;100}$
		$C_{\mathrm{EDP}}$	$C_{E-\mathrm{Mini}\;S\&P\;500}$
		$C_{\mathrm{SABESP}}$	$C_{\mathrm{BB}\;\mathrm{Seguridade}}$
		$C_{\mathrm{Equatorial}\;\mathrm{Energia}}$	$C_{\mathrm{Petrobras}}$
		$C_{\mathrm{EcoRodovias}}$	$C_{\mathrm{JBS}}$
		$C_{\mathrm{Raia}\;\mathrm{Drogasil}}$	$C_{\mathrm{Usiminas}}$
		$C_{\mathrm{Ita}ú\mathrm{sa}}$	$C_{\mathrm{Banco}\;\mathrm{Bradesco}}$
		$C_{\mathrm{Multiplan}}$	$C_{\mathrm{Banco}\;\mathrm{do}\;\mathrm{Brasil}}$
		$C_{\mathrm{Eletrobr}ás}$	$C_{\mathrm{Ita}ú}$
		$C_{\mathrm{Ambev}}$	$C_{B3}$
		$C_{\mathrm{Vale}}$	$C_{\mathrm{Klabin}}$
		$C_{\mathrm{US}\;\mathrm{Dollar}}$	$C_{\mathrm{British}\;\mathrm{Pound}}$
		$C_{\mathrm{Russell}\;2000}$	$C_{\mathrm{Euro}}$
		$C_{\mathrm{NYSE}}$	$C_{\mathrm{Dow}\;\mathrm{Jones}}$
		$C_{\mathrm{Natural}\;\mathrm{Gas}}$	$C_{\mathrm{NYSE}\;\mathrm{AMEX}}$
		$C_{\mathrm{Wheat}}$	$C_{\mathrm{Cboe}\;\mathrm{UK}\;100}$
		$C_{\mathrm{Silver}}$	$C_{\mathrm{Copper}}$
		$C_{\mathrm{Corn}}$	$C_{\mathrm{Oat}}$
		$C_{\mathrm{Soybean}}$	$C_{\mathrm{Rice}}$

C—close serie.

**Table 5 entropy-26-00177-t005:** Data partitioning for each experiment. The specification of the experiment, the start date and the end date considered for the time series, the dates that defined the training and test sets, as well as the appropriate proportions of these sets are presented.

Experiment	Start Date	End Date	Training	Test
E1 - I1	08/19/2013	07/19/2016	08/19/2013–12/18/2015 (80%)	12/19/2015–07/19/2016 (20%)
E1 - I2	04/01/2013	04/01/2017	04/01/2013–03/31/2016 (75%)	04/01/2016–04/01/2017 (25%)
E2 - I1	01/01/2018	01/01/2021	01/01/2018–05/27/2020 (80%)	05/28/2020–01/01/2021 (20%)
E2 - I2	01/01/2018	01/01/2022	01/01/2018–12/30/2020 (75%)	12/31/2020–01/01/2022 (25%)
E3 - I1	01/01/2018	01/01/2021	01/01/2018–05/27/2020 (80%)	05/28/2020–01/01/2021 (20%)
E3 - I2	01/01/2018	01/01/2022	01/01/2018–12/30/2020 (75%)	12/31/2020–01/01/2022 (25%)

**Table 6 entropy-26-00177-t006:** Operators and parameters of the GA for E1.

Operator/Parameter	Value
Population Size	100
Number of Generations	10,000 (filter) | 1000 (wrapper)
Selection	Roulette Wheel
Crossover	two points
Mutation Probability	0.001
Replacement	elitism

**Table 7 entropy-26-00177-t007:** Operators and parameters of the GA for E2 and E3.

Operator/Parameter	Value
Population Size	100
Number of Generations	500
Selection	rank
Crossover	two points
Mutation Probability	0.01
Replacement	elitism

**Table 8 entropy-26-00177-t008:** Hyperparameters evaluated in E1.

Model	Tested Parameters
SVM	Regularization Parameter: 0.1; 0.5; 1; 3; 5; 7; 10
	Degree of the Polynomial Kernel: 1; 2;...; 9
	Gamma Kernel Coefficient: $\frac{1}{n}$; $1/(n*\sigma^{2}\left(x\right));0,1;0,2;...;1,0$
RF	Max Depth: 1; 2;...; 29
	Number of Trees: 10; 20;...; 150
	Number of Features to Consider for the Best Split: 1; 2;...; 9
MLP	Number of Hidden Layers: 2; 3
	Number of Neurons: 2; 4; 8; 16
	Epochs 100; 200;...; 500
KNN	Number of Neighbors: 2; 3;...; 99
	Distance Metric: Manhattan; Euclidean
LR	Regularization: 0.1; 0.5; 1; 3; 5; 7; 10

**Table 9 entropy-26-00177-t009:** Hyperparameters evaluated in E2 and E3.

Model	Tested Parameters
SVM	Regularization Parameter (C): 0.5; 1.0; 1.5; 10.0
	Kernel: Radial Basis, Polynomial
	Gamma Kernel Coefficient (γ): $\frac{1}{n}$; 0.1; 1.0; 10.0
	Degree of the Polynomial Kernel (d): 1, 2, 3, 4
DT	Criterion: Gini, Entropy
	Max Depth: 3, 4, 5, 6, 7
	Minimum Number of Samples to Split: 3, 6, 9
	Minimum Number of Samples to be at a Leaf Node: 1, 4, 7
KNN	Number of Neighbors (k): 1, 2,..., 30
	Weights: Uniform, Distance

**Table 10 entropy-26-00177-t010:** Best results from E1—Bitcoin. Values in bold highlight the best performance for each classifier.

Inteval	Model	Feature Selection	AUROC	Accuracy
1	**ANN**	**wrapper**	**0.5**± 0.03	**50.79%**± 6.9%
	ANN	none	0.5 ± 0.02	50.78% ± 6.99%
	ANN	filter	0.49 ± 0.02	52.5% ± 6.16%
	**KNN**	**filter**	**0.58 ± 0.0**	**59.35% ± 0.0%**
	KNN	wrapper	0.54 ± 0.0	52.8% ± 0.0%
	KNN	none	0.53 ± 0.0	53.74% ± 0.0%
	**LR**	**filter**	**0.54 ± 0.0**	**58.41% ± 0.0%**
	LR	none	0.53 ± 0.0	57.94% ± 0.0%
	LR	wrapper	0.51 ± 0.0	57.94% ± 0.0%
	**RF**	**filter**	**0.51 ± 0.02**	**52.95% ± 3.56%**
	RF	wrapper	0.5 ± 0.01	57.41% ± 0.89%
	RF	none	0.5 ± 0.02	53.77% ± 3.47%
	**SVM**	**filter**	**0.6 ± 0.0**	**59.81% ± 0.0%**
	SVM	wrapper	0.6 ± 0.0	58.88% ± 0.0%
	SVM	none	0.57 ± 0.0	60.75% ± 0.0%
2	**ANN**	**wrapper**	**0.51 ± 0.02**	**57.52% ± 7.88%**
	ANN	filter	0.51 ± 0.03	53.39% ± 9.72%
	ANN	none	0.5 ± 0.02	56.85% ± 7.77%
	**KNN**	**none**	**0.61 ± 0.0**	**55.45% ± 0.0%**
	KNN	filter	0.6 ± 0.0	59.55% ± 0.0%
	KNN	wrapper	0.56 ± 0.0	51.82% ± 0.0%
	**LR**	**wrapper**	**0.57 ± 0.0**	**56.82% ± 0.0%**
	LR	filter	0.53 ± 0.0	62.27% ± 0.0%
	LR	none	0.5 ± 0.0	50.91% ± 0.0%
	**RF**	**none**	**0.51 ± 0.03**	**43.4% ± 5.1%**
	RF	wrapper	0.49 ± 0.03	46.81% ± 6.56%
	RF	filter	0.49 ± 0.03	43.69% ± 5.53%
	**SVM**	**none**	**0.58 ± 0.0**	**65.91% ± 0.0%**
	SVM	filter	0.58 ± 0.0	58.64% ± 0.0%
	SVM	wrapper	0.54 ± 0.0	58.64% ± 0.0%

**Table 11 entropy-26-00177-t011:** Interval 1: Attributes selected by the filter strategy (E1) using SVM.

Day D	Lag (n)	WMA 30 Days
Open	Cost per Transaction (2, 6)	Number of Transactions
	Average Hash Rate (1, 6)	Volume
	Number of Transactions (1, 3)	$C_{\mathrm{Coffee}}$
	Transaction Fees (4, 7)	Open
	Volume (1, 6, 7)	$C_{\mathrm{Platinum}}$
	Direction (1, 5, 7)	$C_{S\&P\;500}$
	High (1, 2)	
	Low (4, 5)	

C—close series.

**Table 12 entropy-26-00177-t012:** Best results from E2—Ibovespa. Values in bold highlight the best performance for each measure.

Interval 1 (I1)			Inteval 2 (I2)		
**Model**	**ACC**	**AUROC**	**Model**	**ACC**	**AUROC**
SVM: 1.5|p|$\frac{1}{n}$|3 (D1—filter)	**59.86**% ± 0.0%	**0.62** ± 0.0	SVM: 1.0|r|$\frac{1}{n}$|0 (D3—wrapper)	56.36% ± 0.0%	0.56 ± 0.0
DT: g|5|4|3 (D2—wrapper)	61.41% ± 1.69%	0.60 ± 0.02	DT: g|4|1|6 (D3—none)	**61.44**% ± 0.0%	**0.61** ± 0.0
KNN: 26|u (D4—none)	55.63% ± 0.0%	0.56 ± 0.0	KNN: 26|d (D3—filter)	55.51% ± 0.0%	0.55 ± 0.0

**Table 13 entropy-26-00177-t013:** Best results from E3—Vale. Values in bold highlight the best performance for each measure.

Inteval 1 (I1)			Interval 2 (I2)		
**Model**	**ACC**	**AUROC**	**Model**	**ACC**	**AUROC**
SVM: 1.5|p|$\frac{1}{n}$|3 (D3—wrapper)	53.52% ± 0.0%	0.52 ± 0.0	SVM: 10.0|r|$\frac{1}{n}$|0 (D3—none)	57.63% ± 0.0%	0.57 ± 0.0
DT: g|5|1|6 (D3—filter)	55.63% ± 0.0%	0.55 ± 0.0	DT: g|3|1|3 (D2—none)	**57.41**% ± 0.69%	**0.58** ± 0.01
KNN: 5|u (D2—wrapper)	**56.34**% ± 0.0%	**0.57** ± 0.0	KNN: 24|u (D3—filter)	53.81% ± 0.0%	0.55 ± 0.0

**Table 14 entropy-26-00177-t014:** Best results by dataset with respect to AUROC measure. The best value for the considered metric in each evaluation scenario is highlighted in bold.

Experiment—Interval	D1	D2	D3	D4
E2—I1	**0.62**	0.60	0.56	0.58
E2—I2	0.50	0.56	**0.61**	0.58
E3—I1	0.51	**0.57**	0.55	0.53
E3—I2	0.50	**0.58**	0.57	0.53

**Table 15 entropy-26-00177-t015:** Best results by feature selection with respect to AUROC measure. The best value for the considered metric in each evaluation scenario is highlighted in bold.

Experiment—Interval	Filter	Wrapper	None
E2—I1	**0.62**	0.60	0.56
E2—I2	0.58	0.56	**0.61**
E3—I1	0.55	**0.57**	**0.57**
E3—I2	0.55	0.57	**0.58**

**Table 16 entropy-26-00177-t016:** Best results by machine learning model with respect to AUROC measure. The best value for the considered metric in each evaluation scenario is highlighted in bold.

Experiment—Interval	KNN	DT	SVM
E2—I1	**0.62**	0.60	0.56
E2—I2	0.56	**0.61**	0.55
E3—I1	0.52	0.55	**0.57**
E3—I2	0.57	**0.58**	0.55

## Data Availability

The data considered for the evaluations conducted in this work are available in the Yahoo Finance, Investing.com, and Nasdaq Data repositories, as mentioned with more details in Section 3.1.

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
