# Peer review of "Genetic Algorithm for Feature Selection Applied to Financial Time Series Monotonicity Prediction: Experimental Cases in Cryptocurrencies and Brazilian Assets"

_entropy, 2024, doi:10.3390/e26030177_

Round 1
Reviewer 1 Report
Comments and Suggestions for Authors
See the attached file.

Comments on the Quality of English LanguageThe authors need to enhance the English in this manuscript.
Author Response
Dear reviewer, we are very grateful for your detailed review. The comments and suggestions have been fully addressed and we believe the manuscript now meets your expectations. In the following, we detail how each of the raised issues has been addressed.
Point 1: The author should thoroughly check the entire manuscript and standardize the naming of various nouns, for example:
(a). In line 102 of the manuscript, the author has abbreviated Genetic Algorithms as GA, but in section 2.2, the author consistently uses the full name. Only in line 140 of this section, the author abbreviates Genetic Algorithms as GA again. The full name is then used in subsequent sections.
(b). In line 84, the author has already abbreviated Principal Component Analysis as PCA, but in line 139, it is abbreviated again.
(c). In line 125, the author marked Support Vector Regression without abbreviation. However, in line 153, the author abbreviates it as SVR. Why wasn’t it abbreviated initially?
(d). In line 141, the author marked artificial neural network, but in line 152, it is marked in uppercase. Since it appears multiple times, I recommend using an abbreviation.
(e). In lines 153, 160, and 247, the author uses “the opening,” “maximum,” “minimum,” and “closing” to denote Open, High, Low, and Close. However, in subsequent Table 3, 4, 11, and Table A1, the author uses Open, High, Low, and Close, etc.
(f). The core of the author’s manuscript is Feature Selection, but in this manuscript, terms like Attribute selection and selection of attribute are used multiple times.
(g). The author uses “Stock” (e.g., line 192) and “Share” (e.g., lines 171 and 241) interchangeably.
(h). In lines 237 and 238, the author has abbreviated Experiment 1 (E1), Experiment 2 (E2), and Experiment 3 (E3). However, in subsequent use, it alternates between the full name and the abbreviation.
(i). In lines 351 and 377, “ROC curve (ROC AUC)” is a strange abbreviation.
Author response: We thank the reviewer for all the observations. In fact, we agree that there was a lack of standardization about the naming conventions in our work, and to maintain the clarity and cohesion of the text, all problems highlighted have been fixed.
Author action: We reviewed and fixed all the issues in the text pointed out by the reviewer. Furthermore, the entire text was deeply revised by the authors and typographical errors and textual standardization problems were corrected.
Point 2: The author’s manuscript has some grammatical errors that need to be checked throughout. For example,
(a). In line 197, “such as, for example, time series of stock prices.”
(b). In Table 9, “Hyperparameters evaluated in experiments E2 e E3.”
Author response: Thank you for raising these issues. The correction of these grammatical errors in the text will definitely improve the text's readability and consequently benefit the readers.
Author action: We reviewed and corrected the issues pointed out by the reviewer.
Point 3: The author uses ACC in the manuscript, as mentioned in line 422: “The metrics are accuracy (ACC).” I cannot understand the specific quantification formula for this metric.
Author response: We thank the author for raising this question. As our work is more specialized in the field of direction forecasting (rise or fall) of the time series, we believe that it makes sense to follow the guidance of Mallqui and Fernandes (2019) who only evaluate metrics related to accuracy, such as accuracy itself and the metric ROC AUC, since the values themselves of the time series are outside the scope of the methodology addressed. However, we agree that this is a very interesting direction to pursue in future work.
Author action: Therefore, we added this intention in the conclusion section. Furthermore, we highlight the information collected here in Section 3.6, which deals with the evaluation methodology considered. To increase understanding of the formulas considered, we added their description and indicated references with their justifications and/or use in scientific works.
Point 4: I recommend the author to include visualizations of the prediction results in the manuscript.
Author response: We thank the reviewer for making this observation. In fact, our work is dedicated to analyzing and detecting the monotonicity of financial time series and not their eigenvalue. Therefore, the proposed framework only detects the upward and downward movement of time series and, therefore, it is not feasible to demonstrate the prediction of time series values using graphical resources.
Author action: We believe that the addition of information that the proposed framework is only responsible for predicting the upward or downward movement of financial time series is effective in highlighting the performance of our contribution and, consequently, highlights the inconvenience in making use of graphical resources to visualization of forecasting results for time series. This addition of details was made in several parts of the text. For example, in the title of the work, in the introduction, in the methodology (Equations 1 and 2), etc.
Reviewer 2 Report
Comments and Suggestions for Authors
The paper empirically tests genetic algorithms for selection of features of classification models that aim to predict next day movement of an asset price. The classification models include neural networks, support vector machines, random forest, logistic regression, or k-nearest neighbor. The goal of the genetic algorithm is to select a subset of a larger set of candidate features in order to optimize performance. The performance measured with accuracy and AUROC show interesting results, i.e. AUROC or ACC significantly above 50%, but not for all the models and feature selection methods. The paper can be recommended for publication with the following minor remarks:
- The feature selection methods should be better described. Even though the notions like Mutual Information, or the Genetic Algorithm parameter meaning could be find in related literature, it should be rather described in the paper, or its appendix, for benefit of the reader.
- The construction of Training/Test split should be better explained or modified. It appears that the Test set is given by a random subset of days in an interval, but this creates an overlap between the training and test sets due to lagged and WMA feature definition. A more consistent way to measure out of sample performance is to use a moving test window preceded by moving training window, which would be consistent with a practical application of a prediction method.
- The performance of the models is shown in terms of ACC and AUROC, but a standard question is whether the prediction method is good enough to generate systematic profitability including transaction coats of the corresponding daily long-short strategy. Some results of the best strategies should be reported.
- The captions of tables should provide more details so that the reader does not have to search the relevant information in the text. For example, Table 11 should specify the models and Tables 14-16 the measure.
Author Response
Dear reviewer, we are very grateful for your detailed review. The comments and suggestions have been fully addressed and we believe the manuscript now meets your expectations. In the following, we detail how each of the raised issues has been addressed.
Point 1: The feature selection methods should be better described. Even though the notions like Mutual Information, or the Genetic Algorithm parameter meaning could be find in related literature, it should be rather described in the paper, or its appendix, for benefit of the reader.
Author response: Thank you for the interesting observations. We now realize that those methods were not described in the paper. And we fully agree that these descriptions are important to improve the reading understanding and should be provided.
Author action: We have added explanations about the concepts of mutual information, as well as the other metrics such as Pearson correlation and F-Statistic in Section 3.4.1. Furthermore, we included better descriptions for the Genetic Algorithm parameters in Section 3.4.2.
Point 2: The construction of Training/Test split should be better explained or modified. It appears that the Test set is given by a random subset of days in an interval, but this creates an overlap between the training and test sets due to lagged and WMA feature definition. A more consistent way to measure out of sample performance is to use a moving test window preceded by moving training window, which would be consistent with a practical application of a prediction method.
Author response: We thank the reviewer for raising this problem. In fact, we understand that the train/test split methodology should be better explained to avoid misunderstandings. We understand that greater details should be provided on the choice of methodology used to partition the data. We split the dataset into training and testing by dividing the data into two subsets, keeping the time series in chronological order. The first subset was used for training and the second for testing.
Author action: Therefore, we wrote a paragraph explaining the nature of dependence between sequential values of time series and detailing how the data set should be divided between training and testing in subsection 3.3. Furthermore, we have added the data periods used for training and testing in Table 5.
Point 3: The performance of the models is shown in terms of ACC and AUROC, but a standard question is whether the prediction method is good enough to generate systematic profitability including transaction coats of the corresponding daily long-short strategy. Some results of the best strategies should be reported.
Author response: We thank the author for raising this consideration. The systems we proposed are for financial time series monotonicity prediction. This type of prediction is very useful to assist an investor’s daily decision-making in order to serve as an indication for the eventual purchase or sale of a credit share. However, the main goal of our proposed methodology is to evaluate the use of feature selection methods based on meta-heuristic. Thus, the application of our contributions in a real trading scenario may be better explored in future research.
Author action: We highlighted the benefits that the use of a prediction framework, such as the one we proposed, and how it can assist the investor’s decision-making, in the Introduction section.
Point 4: The captions of tables should provide more details so that the reader does not have to search the relevant information in the text. For example, Table 11 should specify the models and Tables 14-16 the measure.
Author response: We appreciate the reviewer raising this issue.
Author action: We added the requested details in the captions of the mentioned tables and other tables also had their captions increased with greater details.
Reviewer 3 Report
Comments and Suggestions for Authors
You propose using TS methods to forecast the daily price of Bitcoin and the index Ibovespa, and Vale Shares. In the development you set two systems, and carry a performance analysis based on feature selection method and the combination of other ML systems. Several question need an answer.
1. The pre-processing of data can have an influence in the final results. Could you tell about it some more?
2. When treating structural missing data, you indicate that these are replaced by the last available figure. Why some imputation method has not been used?
3. How many missing data are in each series? Are all missing data treated in the same way for different series?
4. In table 5 the data partitioning is presented. For example, in experiment E1-I1, you use 3 years of daily data, 4 years for E1-I2, and so on; Why these differences?
5. How you select the training set? The first 80% or 75% of the time series, and the last 20%, 25% of each? These intervals should be presented.
6. With 3/4 years of data, and if there are more than a hundred attributes, so, it is necessary to include how many parameters you have used in each model.
7. You talk about several goodness of fit measures, but it seems that only ROC AUC ones are used. Could you confirm?
8. If the objective is to forecast some series, such as Bitcoin price and other, at the end you should have a forecast for a certain time period, such as a day, or an interval of days. But, if you have used, say 3 years to estimate the model as training set, and the following year as a test set, how can you forecast a full year of prices? This is not possible. And how have you selected the training set? Put forward an example.
9. It would be informative to present other goodness of fit measures, and a Theil decomposition of the mean square error of prediction, not only the ROC.
Author Response
Dear reviewer, we are very grateful for your detailed review. The comments and suggestions have been fully addressed and we believe the manuscript now meets your expectations. In the following, we detail how each of the raised issues has been addressed.
Point 1: (1) The pre-processing of data can have an influence in the final results. Could you tell about it some more?
Author response: We are very grateful for this note and we fully agree with the reviewer. In this way, we revised the text of the time series preprocessing section and we have added more details about the adopted methodology.
Author action: We have added information and details in preprocessing Section 3.2.
Point 2: (2) When treating structural missing data, you indicate that these are replaced by the last available figure. Why some imputation method has not been used?
Author response: We thank the reviewer for raising this question. We agree that the issue should be better explained in the text. In this case, the choice of replacing missing values with the last valid value in the time series was taken due to its simplicity and the characteristics of the implemented systems. The Bitcoin system does not need sophisticated imputation considering that series with missing data are only used after being transformed into the 30-day WMA and represent only a small portion of the system's total input series. In the case of the second system, referring to the Ibovespa and Vale series, again, there are a few series that present missing values, mainly because the series used are, in most cases, from the Brazilian Stock Exchange itself, B3. As there are very few missing values, a more sophisticated imputation method would hardly yield much better results, so we opted for this simpler method. Therefore, we added all this information to the text in order to make this note clearer.
Author action: We added details about the choice of methodology for filling in missing data in the penultimate paragraph of Section 3.2.
Point 3: (3) How many missing data are in each series? Are all missing data treated in the same way for different series?
Author response: We thank the reviewer for this observation. We understand that these details should be stated more clearly. We have different amounts of missing data between the datasets used in the experiments. Furthermore, the same technique was used to deal with this problem in all of them.
Author action: Therefore, we added the details of the amounts of missing data in Section 3.2. This was done by adding text in the penultimate paragraph of the section. Furthermore, we created a new subsection (4.1) in the results section to highlight exactly the portion of missing data in each time series considered.
Point 4: (4) In table 5 the data partitioning is presented. For example, in experiment E1-I1, you use 3 years of daily data, 4 years for E1-I2, and so on; Why these differences?
Author response: We thank the reviewer for this interesting question. The prediction systems proposed in our paper are an adaptation of several points from the study by Mallqui and Fernandes [10]. For comparison reasons, we kept our first experiment attached to these periods. Besides, we proposed two systems, which both have parts based on the study of Mallqui and Fernandes. For comparison reasons, we have used the same period presented in that study for Experiment 1 (Bitcoin). For both Experiments 2 and 3, we used the same duration with more recent data. We understand that this choice should be better explained in the text. For this reason, we expand this justification in the section that deals with data partitioning (Section 3.3).
Author action: Addition of text with the justification for the methodological choice in data partitioning at the end of the paragraph in Section 3.3.
Point 5: (5) How you select the training set? The first 80% or 75% of the time series, and the last 20%, 25% of each? These intervals should be presented.
Author response: We thank the reviewer for the note. In the same way, as we proceeded in the previous note, we understand that greater details should be provided on the choice of methodology used to partition the data. We split the dataset into training and testing by dividing the data into two subsets, keeping the time series in chronological order. The first subset was used for training and the second for testing.
Author action: Therefore, we wrote a paragraph explaining the nature of dependence between sequential values of time series and detailing how the data set should be divided between training and testing in subsection 3.3. Furthermore, we have added the data periods used for training and testing in Table 5.
# Point 6: (6) With 3/4 years of data, and if there are more than a hundred attributes, so, it is necessary to include how many parameters you have used in each model.
Author response: We thank the reviewer for raising this question. It is worth highlighting that the intention of this work is precisely to optimally select the set of temporal variables that accompany the main financial series established in each of the experiments considered in order to reduce the prediction error for future values of these series. Therefore, the complete space of characteristics that we will consider is formed by the series in Table 3, 4 and A1 of the appendix, but our objective is to select a partition from this set in order to enhance the performance of the models considered. Therefore, in each execution of our framework, we will have a subset of characteristics considered.
Author action: We have added an explanation of this intention at the end of Section 3.1.
Point 7: (7) You talk about several goodness of fit measures, but it seems that only ROC AUC ones are used. Could you confirm?
Author response: Thank you very much for raising this consideration. It is important to note that, although the GA fitness function is defined based on the ROC AUC metric, we also consider the accuracy of the model in our experiments.
Author action: We have added this explanation to the end of Section 3.4.1.
Point 8: (8) If the objective is to forecast some series, such as Bitcoin price and other, at the end you should have a forecast for a certain time period, such as a day, or an interval of days. But, if you have used, say 3 years to estimate the model as training set, and the following year as a test set, how can you forecast a full year of prices? This is not possible. And how have you selected the training set? Put forward an example.
Author response: Thank you for raising this issue. In the training set the data refer to the initial period of the interval and in the test set they refer to the final period, and they do not overlap. Regarding your question about the very long prediction period, in the proposed systems, predictions are not made on a recurring basis, as is common in ARIMA models, for example. We make the predictions day by day, so to predict the price direction of a specific day in the series the data available from the previous day was used.
Author action: We believe that the addition of information that the proposed framework is only responsible for predicting the upward or downward movement of financial time series is effective in highlighting the performance of our contribution. Therefore, we included a mathematical formulation of the proposed forecasting methodology at the beginning of Section 3 to highlight the objective of this framework and avoid possible misunderstandings. Furthermore, about the training set selection, we wrote a paragraph detailing how the data set should be divided between training and testing in subsection 3.3. As an example of this selection process, consider a sequence of 100 days of data, from which the data relative to the first 75 days is used to fit the models, and the remaining 25 days to the test.
Point 9: (9) It would be informative to present other goodness of fit measures, and a Theil decomposition of the mean square error of prediction, not only the ROC.
Author response: We thank the author for raising this question. As our work is more specialized in the field of direction forecasting (rise or fall) of the time series, we believe that it makes sense to follow the guidance of Mallqui and Fernandes (2019) who only evaluate metrics related to accuracy, such as accuracy itself and the metric ROC AUC, since the values themselves of the time series are outside the scope of the methodology addressed. However, we agree that this is a very interesting direction to pursue in future work.
Author action: Therefore, we added this intention in the conclusion section. Furthermore, we highlight the information collected here in Section 3.6, which deals with the evaluation methodology considered.
Round 2
Reviewer 1 Report
Comments and Suggestions for Authors
I recommend this article for publication in this journal.
Reviewer 3 Report
Comments and Suggestions for Authors
I think you have presented the methodology and results clearer in this revised version. Maybe some well known concepts, as the definition of Pearson correlation could be omitted.